# Performance of Radiomics in Microvascular Invasion Risk Stratification and Prognostic Assessment in Hepatocellular Carcinoma: A Meta-Analysis

**DOI:** 10.3390/cancers15030743

**Published:** 2023-01-25

**Authors:** Sylvain Bodard, Yan Liu, Sylvain Guinebert, Yousra Kherabi, Tarik Asselah

**Affiliations:** 1Service de Radiologie Adulte, Hôpital Universitaire Necker-Enfants Malades, AP-HP Centre, 75015 Paris, France; 2Faculté de Médecine, Université Paris Cité, 75007 Paris, France; 3CNRS, INSERM, UMR 7371, Laboratoire d’Imagerie Biomédicale, Sorbonne Université, 75006 Paris, France; 4Faculty of Life Science and Medicine, King’s College London, London WC2R 2LS, UK; 5Median Technologies, 1800 Route des Crêtes, 06560 Valbonne, France; 6Service d’Hépatologie, INSERM, UMR1149, Hôpital Beaujon, AP-HP.Nord, 92110 Clichy, France

**Keywords:** hepatocellular carcinoma, imaging phenomics, radiomics, risk stratification and prognostication

## Abstract

**Simple Summary:**

The poor prognosis of advanced hepatocellular carcinoma (HCC) warrants a personalized approach. Our objective was to assess the value of imaging phenomics for risk stratification and prognostication of HCC. We performed a meta-analysis of manuscripts published to January 2023 on MEDLINE and showed that imaging phenomics is an effective solution to predict prognosis or treatment response in patients with HCC.

**Abstract:**

Background: Primary liver cancer is the sixth most commonly diagnosed cancer and the third leading cause of cancer death. Advances in phenomenal imaging are paving the way for application in diagnosis and research. The poor prognosis of advanced HCC warrants a personalized approach. The objective was to assess the value of imaging phenomics for risk stratification and prognostication of HCC. Methods: We performed a meta-analysis of manuscripts published to January 2023 on MEDLINE addressing the value of imaging phenomics for HCC risk stratification and prognostication. Publication information for each were collected using a standardized data extraction form. Results: Twenty-seven articles were analyzed. Our study shows the importance of imaging phenomics in HCC MVI prediction. When the training and validation datasets were analyzed separately by the random-effects model, in the training datasets, radiomics had good MVI prediction (AUC of 0.81 (95% CI 0.76–0.86)). Similar results were found in the validation datasets (AUC of 0.79 (95% CI 0.72–0.85)). Using the fixed effects model, the mean AUC of all datasets was 0.80 (95% CI 0.76–0.84). Conclusions: Imaging phenomics is an effective solution to predict microvascular invasion risk, prognosis, and treatment response in patients with HCC.

## 1. Introduction

Primary liver cancer is the sixth most commonly diagnosed cancer and the third leading cause of cancer death worldwide in 2020, with approximately 906,000 new cases and 830,000 deaths [1]. Hepatocellular carcinoma (HCC) accounts for 75%–85% of cases [1], and incidence rates continue to increase rapidly, by about 3% per year in women and 4% per year in men [2]. It is the fourth most common malignancy and the third leading cause of tumor-related death in China [3]. Due to its highly aggressive nature, HCC is one of the deadliest primary cancers, with a 5-year survival rate of 10% or less [4].

The poor prognosis of advanced HCC warrants a personalized approach that advances in the discovery of new surrogate biomarkers. Therefore, there is an unmet need to develop novel, reliable tools that can accurately diagnose tumors at an early stage, predict patient prognosis, and dynamically monitor treatment response, in order to improve the clinical outcome of HCC. The tools should also be available and affordable for patients and the health care system. The development of imaging phenomics as a surrogate for genomic and transcriptomic analysis might be one of the most appropriate solutions with minimal new investment. This could revolutionize HCC diagnosis, patient stratification, and personalized treatment.

In this regard, the objective of this article was to assess the value of imaging phenomics for microvascular invasion risk stratification and prognostication of HCC.

## 2. Materials and Methods

We performed a meta-analysis of studies on the prognostic value of imaging phenomics in microvascular invasion risk stratification and prognostication of hepatocellular carcinoma published to January 2023. This meta-analysis follows Preferred Reporting Items for Systematic reviews and Meta-Analyses (PRISMA) 2020 recommendations and is registered under the number CRD42023386763 (PROSPERO).

### 2.1. Search Strategy

On 1 January 2023, the PubMed (MEDLINE) database was queried to identify potentially relevant articles using the below search strategy and inclusion/exclusion criteria (Table 1). We chose to focus the study on this database because of the many high-quality articles referenced and easily accessible. Two reviewers screened each record and each report. We also screened the reference lists of the included studies, and relevant systematic reviews and meta-analyses. All of the records from these searches were imported via EndNote into Rayyan (a free web application for screening abstracts) to proceed with the screening.

### 2.2. Identification of Relevant Published Studies

We identified all studies that assessed the prognostic value of imaging phenomics in hepatocellular carcinoma. Studies were eligible for inclusion if they (1) were retrospective or prospective; (2) assessed imaging phenomics; (3) reported pathological analysis of tumors including MVI information; and (4) reported the association with cancer prognosis or response to treatment. However, it was restricted to HCC, and no other disease indications were investigated. In addition, the following exclusion criteria were applied: only abstracts available, studies not in the English language, non-human studies, diagnostic studies, editorial style reviews, abstracts and posters, conference papers, case reports, and studies with high-risk populations susceptible to having HCC but without overt HCC.

Two reviewers examined each title, keywords, and abstract and then selected full-text articles according to the pre-specified eligibility criteria.

### 2.3. Data Extraction

A predesigned data collection form was prepared to extract the relevant information from the selected studies including study location (defined as Eastern Asia, Western Europe, North America, and North Africa; location of international RCTs was determined according to the primary investigator); number of involved centers (either single or multicenter); presence of a university-affiliated center; funding source (profit, non-profit, or mixed); date of journal publication; type of journal (i.e., radiological or clinical), median journal impact factor in the 2 years before publication, the first author’s name, study objectives (including essential information such as the aim of the study and the characteristics of the study population), sample size, and performance. Furthermore, for each study, the method (e.g., tissue type), directions of the associations, and, when possible, the reported measures of associations (e.g., correlation analysis, hazard ratio (HR), odds ratio (OR), receiver operating characteristics (ROC), and confidence interval (CI) were reported. When applicable, effect sizes were included. Data extraction was performed independently by two seniors. To control its quality, the results were compared, and disagreements were resolved by a third senior.

### 2.4. Outcome Assessment and Statistical Methods

Regarding the possible heterogeneity of the studies, we sought to pool the results using a random-effects meta-analysis model. However, it was not feasible to pool all the data due to limited data, differences in exposure and outcomes, and input parameters. Finally, we selected studies from imaging phenotyping subgroups that disclosed the univariate impact of patient outcome, MVI (microvascular invasion), with a 95% CI, and organized forest plots to quantify the importance of imaging phenomics in HCC MVI prediction.

Meta-analysis was carried out using the methodology part of Review Manager 5.3, Cochrane’s meta-analysis software [5], and R statistical software with package Metafor [6]. The subgroup statistical analysis is summarized in the below steps:

#### 2.4.1. Assessment of Risk of Bias

Since the included studies were observational cohort studies of prognostic factors, the QUIPS (Quality in Prognosis Studies) tool was used [7]. It allows for quality assessment in six domains: study participation, study attrition, prognostic factor measurement, outcome measurement, adjustment for other prognostic factors, and statistical analysis/reporting.

#### 2.4.2. Data Synthesis and Analysis

The outcome measure for the meta-analysis was incident MVI in individuals compared to non-MVI. The effect measures reported in the included studies were area under the curve (AUC). The results were pooled, and an overall estimate of AUC was obtained using a mixed-effects model. It is necessary to consider the study heterogeneity during data extraction and the statistical heterogeneity measured by the I2 statistic. Publication bias was evaluated using the visual inspection of funnel plots. The prognostic factor with sufficient data and homogeneity between studies to carry out the meta-analysis was radiomics signatures extracted from the liver after the injection of a contrast agent.

#### 2.4.3. Assessment of Publication Bias

Data preparation: For all of the presented studies, the following parameters were included in the analysis to assess the potential bias (author, year, country). For the parameter of interest: (1) the estimation of the effect AUC; (2) sample size; and (3) CI were integrated into the raw data. Since the random effect modeling of the parameter of interest requires the standard error, this was computed from the equation below:CI_upper_ = AUC + t*SE
where CI_upper_ is retrieved from the relevant publication and t = 1.92. In each study, training and independent validation cohorts were separated and considered different, but shared the same study parameters (author, year, country) for exploring the potential source of publication bias.

#### 2.4.4. Modeling

Mixed-effects modeling: Differences in the methods and sample characteristics may introduce variability (“heterogeneity”) among the true effects. One way to model the heterogeneity is to treat it as purely random. This leads to the random-effects model, given by θi = μ + ui, where ui ∼ N (0, τ2). Therefore, the true effects are assumed to be normally distributed with mean μ and variance τ2. The goal is then to estimate μ, the average true effect, and τ2, the amount of heterogeneity among the true effects. Alternatively, we can include one or more moderators (study-level variables such as author, year, and country) in the model that may account for at least part of the heterogeneity in the true effects. This leads to the mixed-effects model, where the analysis aims to examine to what extent the moderators included in the model influence the size of the average true effect.

## 3. Results

We identified 362 articles via a PubMed database search. A summary of the study selection process is summarized in Figure 1. Finally, 36 studies were included following the above inclusion criteria, addressed imaging phenomics, and HCC prognosis to predict the prognosis and response to the treatment of HCC.

### 3.1. Description of Radiomics Prediction

For each eligible study, a detailed description of radiomics prediction of patient outcome (Table 2 and Table 3) and its prediction of underlying pathology (Table 4, Table 5 and Table 6) was reported. It included the study objectives, sample size (both on the training and validation sets), and their performance.

### 3.2. Subgroup Analysis on Radiomics Prediction of MVI

MVI is defined as the presence of micrometastatic HCC emboli within the vessels of the liver, and is a critical determinant of early recurrence and survival [44]. According to bias evaluation, seven studies were extracted from imaging phenotyping groups that had disclosed the univariate impact of MVI measured by histopathology. The evaluation of bias was performed. For mixed-effects models (i.e., models with cohort type as moderator), the plot shows the individual residual post modeling on the *x*-axis against the corresponding standard errors (Figure 2A) and sampling variance or standard error (Figure 2A). A vertical line indicates the estimate based on the model. A pseudo confidence interval region was drawn around this value with bounds equal to ±1.96 SE, where SE is the standard error value from the *y*-axis (assuming level = 95). The two graphics clearly showed a potential study bias (non-symmetrical funnel plots) that would violate assumptions for fixed and random effect modeling strategies. Taking this into consideration, a mixed effect modeling was more appropriate.

According to the influence diagnostics, Figure 3A shows a plot of the externally standardized residuals as a function of each of the studies. Highlighted is the potential influence point. Figure 3B shows a plot of the leave-one-out estimates of the amount of heterogeneity (a) and leave-one-out values of the test statistics for heterogeneity (a) as a function of each of the studies. Highlighted in red is the potential influence point (study). The point-of-influence analysis shows that there is one study (Yao et al. 2018) that is potentially an outlier; however, the study needs to be looked at more closely to analyze the characteristics of the cohort and see if we do have a subgroup of patients, and whether it is potentially interesting to represent them with more studies specifically selected with the characteristics of this cohort. 

NB. The presence of outliers is highlighted in each exploratory analysis graphic. Removal of these outlier studies would reduce the amount of heterogeneity and increase the precision of the estimated average outcome. However, before removing those studies, an investigation is needed to determine the reason for the unusual results. Outliers and influential cases can reveal patterns that may lead to new insights about the study characteristics that could act as potential moderators.

Finally, Figure 4 shows QQ (quantile–quantile) plots for the effect normality assumption.

NB. Normal QQ are useful in meta-analyses to check various aspects and assumptions of the data. Ideally, the points in the plot should fall on a diagonal line with a slope of 1, going through the (0,0) point. Deviations from this may indicate that (1) the (residual) heterogeneity in the actual effects is non-normally distributed, (2) there are subgroups in the data (that are not adequately modeled by any moderators already included in the model), and/or (3) that publication bias is present.

Forest plots were demonstrated (Figure 5) to quantify the importance of imaging phenomics in HCC MVI prediction. When the training and validation datasets were analyzed separately by the random-effects model, it showed that radiomics had good MVI prediction in the training datasets with a mean AUC of 0.81 (95% CI 0.76–0.86). Similar results were found in the validation datasets, having a mean AUC of 0.79 (95% CI 0.72–0.85). Using the fixed effects model, the mean AUC of all datasets was 0.80 (95% CI 0.76–0.84).

## 4. Discussion

Our study shows the importance of imaging phenomics in HCC MVI prediction. When the training and validation datasets were analyzed separately by the random effects model, in the training datasets, radiomics had good MVI prediction (AUC of 0.81 (95% CI 0.76–0.86)). Similar results were found in the validation datasets (AUC of 0.79 (95% CI 0.72–0.85)). Using the fixed effects model, the mean AUC of all datasets was 0.80 (95% CI 0.76–0.84). In the literature, radiomics has produced encouraging results associated with underlying histopathology [13,32,33,34,35,36,37,40,41,42,43] or accurately predicted patient clinical outcome [9,10,11,12,13,14,15,16,17,18,24,26,30]. Although this method is still in its infancy, its unique ability to examine a tumor, its surrounding tissue, and liver parenchyma as a whole allows for an intra-tumoral heterogeneity and extra-tumor microenvironment to be observed, which permits radiology to move beyond the tumor size to other “hidden” features to be discerned with quantitative approaches. Many radiomics studies have focused on the relationship between imaging features and clinical characteristics including recurrence, treatment response after therapeutic agents, and the survival of people with HCC. Shan and colleagues performed feature analysis on the tumor lesion for tumoral radiomics (T-RO) and the peritumoral area for peritumoral radiomics (PT-RO) with pretreatment multiphase liver CT images. They found that the CT-based PT-RO model effectively predicted the early recurrence of HCC compared with the T-RO model and the conventional imaging feature peritumoral enhancement [9]. Other radiomics features based on CT texture have also been reported, suggesting the value of radiomics as a surrogate to predict the patients’ early recurrence and survival after curative therapy including ablation, volumetric modulated arc therapy, surgical resection, and liver transplantation [9,10,11,12,13,14,15,16,18]. For the intermediate stage, a study of 88 patients treated with TACE found that the radiomics approach combined with clinical characteristics could effectively predict the patients’ survival (HR of 19.88, *p* < 0.0001) [17]. These findings might be explained with CT radiomics studies, which demonstrated the accurate diagnosis of the presence of MVI on the pathologic specimen [13,33,35,36,37,45].

Recently, Liujun et al. showed, in a meta-analysis of twenty-two studies with 4129 patients, that radiomics was a promising noninvasive method that has high preoperative diagnostic performance for MVI status [46]. The pooled sensitivity, specificity, and area under the receiver AUC were 84% (95% CI: 81, 87), 83% (95% CI: 78, 87), and 0.90 (95% CI: 0.87, 0.92). These results are consistent with Lv et al. [45] and Zhong et al. [47], who found in a systematic review and meta-analysis an AUC of 0.85 (95% CI 0.82–0.89), 0.87 (95% CI 0.83–0.92), and 0.74 (95% CI 0.67–0.80) for the CT, MR, and ultrasound radiomics models, respectively.

Radiomics scores that could predict the infiltration of tumor-infiltrating CD8⁺ T cells were developed (AUC: training set 0.75, 95% [CI] 0.66–0.85; validation set 0.71, 95% [CI] 0.55–0.86) [32], which might be useful in identifying potential people with HCC who can benefit from immunotherapies, although the score should be further validated in large-scale prospective cohorts. The first radio-genomic study of HCC performed in 2007 [48] combined 28 imaging features and reconstructed gene models, representing 78% of the global gene expression profiles, revealing cell proliferation, liver synthetic function, and patient prognosis. Correlating specific imaging features with underlying genotypes might allow imaging to be used as a surrogate for expression profiling or genome sequencing when tissue samples are unavailable. Another study showed that some imaging traits (infiltrative pattern and macrovascular invasion) were associated with proliferative signatures and the CK19 signature [49].

Compared to CT imaging, the signal intensity of MRI is not easy to digitize or standardized for radiomic analyses. Nevertheless, several MRI-based studies are emerging, and features derived from non-contrast MRI sequences could differentiate HCC pathological grading [38,39,41]. In addition, a strong association between imaging features and immune score on hepatobiliary phase imaging (combined radiomics-based clinical model, AUC, 0.93 (95% [CI] 0.88–0.97)) was reported, and both the validation cohort and calibration curves showed good agreement [40]. However, MR imaging is more vulnerable to imaging artifacts such as motion and magnetic susceptibility, and it is challenging in liver disease. Thus, more efforts are needed to improve the reproducibility between different scanners and repeatability, even with the same scanner.

Furthermore, radiomics feature analysis is a quantitative method that could assess the tumor heterogeneity or liver parenchymal changes by exploring the distribution and connection of pixel gray levels in the CT image or signal in the MRI. The conventional radiomics workflow involves imaging acquisition, delineation, and segmentation of the region of interest, the extraction of imaging features, mining data, and developing models to associate with the underlying pathology or clinical outcomes. However, due to the technical steps, there are remaining issues for clinical use and some common limitations such as intra-observer and inter-observer variability, quality influenced by slice thickness and machine acquisition parameters, some studies using only arterial phase and some using multiphase imaging, single-center retrospective study, small sample size, or biases related to the treatment method. As listed, first, variability is a critical issue related to many factors such as the imaging acquisition protocol, method of segmentation, method for extracting imaging features, and acquisition of clinical and pathological data. Moreover, the intra- and inter-observer variability when radiologists delineate the liver lesion remains an unresolved issue, as it is still challenging and time-consuming due to the indistinct border of the tumors. Therefore, standardization and automatic segmentation in radiomics are the keys if these methods gain broad adoption in clinical use. Second, many radiomic studies are retrospective, with a small sample size. Therefore, it is important to have independent validation with existing cohorts using larger datasets and “big data”, or prospective validation studies with a pre-defined cut-off and statistical power. Third, ideally, a prospective study design is preferable with harmonized treatment methods and was conducted in a multicentral setting, which helps patient recruitment but at the cost of increased protocols and data processing variability.

The prediction interval and distribution (0.80 (95% CI 0.76–0.84)) derived from the analysis were acceptable in our study. However, our study was limited to the MEDLINE database, which may be a limitation, although it offers many quality articles.

HCC demonstrates a complex genetic and epigenetic landscape. Therefore, drug targeting a single aberrant pathway might not be adequate to regress tumor growth. Even targeting multiple pathways like sorafenib, a multikinase inhibitor of Raf, VEGFR, and platelet-derived growth factor receptor-β, only increased the median overall survival by two months in advanced HCC [50]. It took nearly ten years until the portfolio of effective drugs finally expanded to cancer immunotherapy, and recent studies such as the positive phase III study (IMbrave150) with the combination of atezolizumab (anti-PD-L1) and bevacizumab (anti-angiogenesis) open a new era in the treatment of this deadly disease [51,52]. Radiomic technology could play a key role and provide insights into a vast number of potential targets for molecular targeted therapy, which helps to understand tumor biology and assist clinicians in selecting the right therapeutic agents and evaluating the treatment response earlier than the “wait and see if it shrinks” approach currently employed.

In turn, this underscores the urgent need for assembling large, curated clinical and image data registries and robust AI methods that will reliably predict the diagnosis and outcome to guide therapy based on specific HCC subtypes.

## 5. Conclusions

Imaging phenomics is an effective solution to predict microvascular invasion risk, prognosis, and treatment response in patients with HCC.

## Figures and Tables

**Figure 1 cancers-15-00743-f001:**
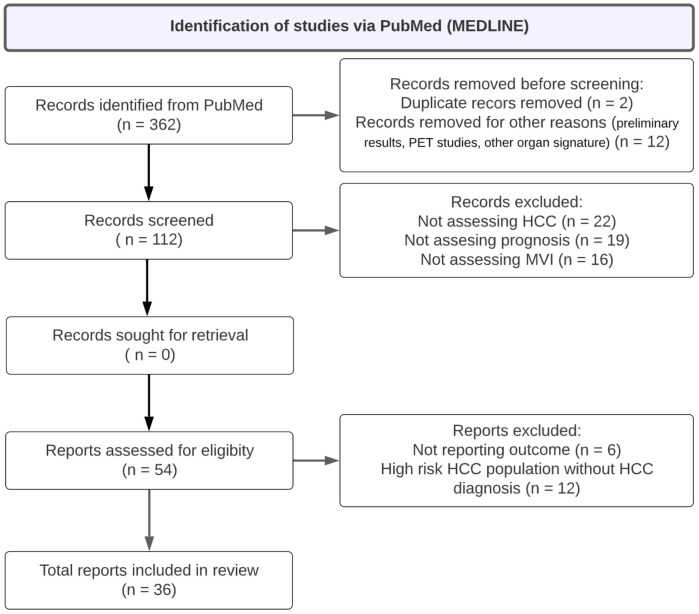
Flow diagram of the study selection process.

**Figure 2 cancers-15-00743-f002:**
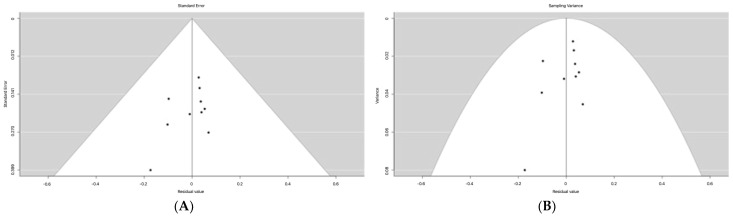
Funnel plots: Individual residual post-modeling on the *x*-axis against the corresponding standard errors (*y*-axis, in decreasing order) (**A**) and sampling variance or standard error (**B**). NB. In the absence of publication bias and heterogeneity, the points form a funnel shape, with the majority of the points falling inside of the pseudo-confidence region with bounds θ ± 1.96SE, where θ is the estimated effect or outcome based on the fixed-effects model and SE is the standard error value from the *y*-axis.

**Figure 3 cancers-15-00743-f003:**
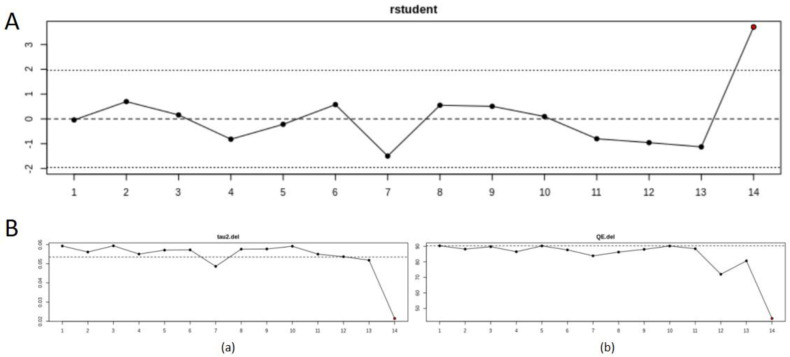
Plot of the influence diagnostics. (**A**) Plot of the externally standardized residuals as a function of each of the studies. Potential influence point. (**B**) Plot of the leave-one-out estimates of the amount of heterogeneity (**a**) and leave-one-out values of the test statistics for heterogeneity (**b**) as a function of each of the studies (optimal value = 100). Potential influence point.

**Figure 4 cancers-15-00743-f004:**
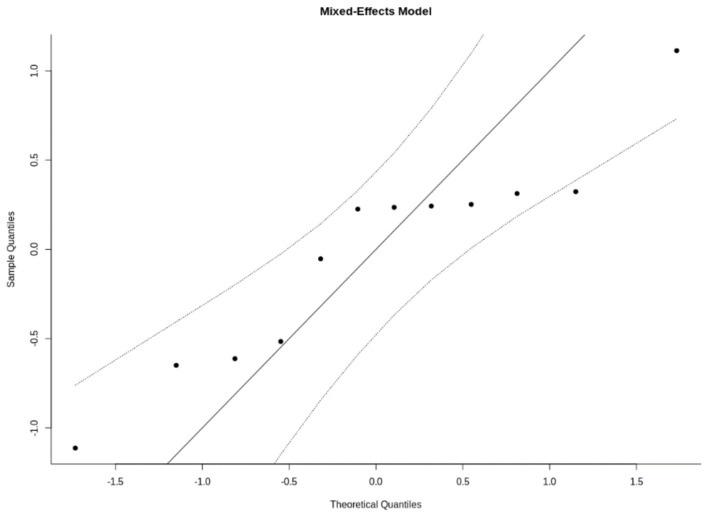
QQ plots for the effect normality assumption (mixed-effects models).

**Figure 5 cancers-15-00743-f005:**
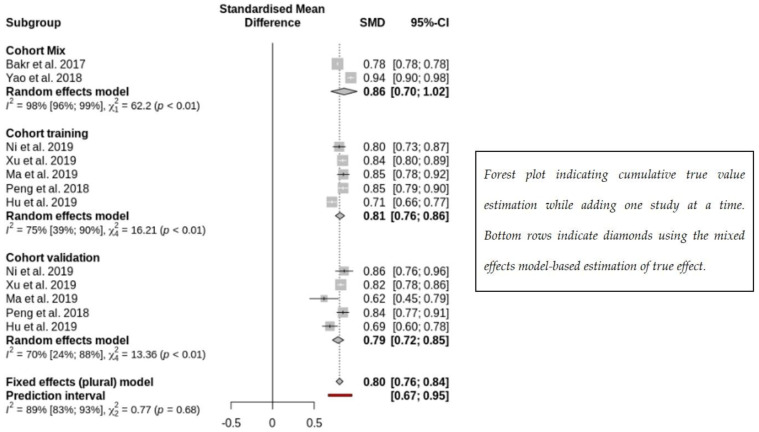
Radiomics prediction of microvascular invasion using mixed-effects modeling [13,33,35,36,37,42,43].

**Table 1 cancers-15-00743-t001:** Search terms.

Search keywords	(imaging data extraction OR radiomics AND hepatocellular carcinoma AND microvascular invasion) NOT ([animals]/lim NOT [humans]/lim) NOT ([Conference Abstract]/lim OR [Letter]/lim OR [Note]/lim OR [Editorial]/lim)
Period	To January 2023

**Table 2 cancers-15-00743-t002:** Performance of radiomics approaches (using CT features to predict patient outcome).

Author/Year	Study Objectives	Training Set Sample Size	Validation Set Sample Size		Performance (Training Set)	Performance (Validation Set)
					[95%CI]	[95%CI]
Wei et al. 2021 [8]	To identify a new radiomics signature using imaging phenotypes and clinical variables for risk prediction of OS after stereotactic body radiation therapy.	167 data were split into training (75% of 4-folds), validation (25% of 4-folds) and testing fold (1-fold)			c-indices nested cross-validation scheme:	
					- radiomics: 0.579 (95%CI: 0.544–0.621)	
					- clinical: 0.629 (95%CI: 0.601–0.643)	
					- image input: 0.581 (95%CI: 0.553–0.613)	
					- combined models: 0.650 (95%CI: 0.635–0.683)	
Shan et al. 2019 [9]	To predict early recurrence after surgical or ablation.	109	47		PT-RO: AUC 0.80 [0.72, 0.89]	PT-RO: AUC 0.79 [0.66, 0.92]
					T-RO: AUC 0.82 [0.74, 0.90]	T-RO: AUC 0.62 [0.46, 0.79]
					PT-E: AUC 0.64 [0.56, 0.72]	PT-E: AUC 0.61 [0.47, 0.74]
Yuan et al. 2019 [10]	To predict early recurrence after curative ablation.	129	55		Portal venous phase model + clinicopathological factors.	Portal venous phase model + clinicopathological factors.
					C-index: 0.792 [0.727–0.857]	C-index: 0.755 [0.651–0.860]
Guo et al. 2019 [11]	To predict recurrence of HCC after liver transplantation.	93	40		C-index of 0.785 [0.674–0.895]	C-index of 0.789 [0.620–0.957]
Ning et al. 2019 [12]	To predict early recurrence (at least 1-year FU).	225	100		AUC: 0.818 [0.760–0.865]	AUC: 0.719 [0.621–0.805]
Xu et al. 2019 [13]	To predict PFS and OS.	495	-		OR: 2.34	
					Median PFS: 49.5 vs. 12.9 months; median OS: 76.3 vs. 47.3 months	
Cai et al. 2019 [14]	To predict post-hepatectomy liver failure.	80	32		AUC: 0.822 [0.726–0.917]	AUC: 0.762 [0.576–0.948]
Akai et al. 2018 [15]	To predict random survival forest.	127		-	Predicted individual risk (*P* = 1.1 × 10^−4^ for DFS, 4.8 × 10^−7^ for OS).	
					The only unfavorable prognostic factors were high predicted risk (HR = 1.06 per 1% increase, *P* = 8.4 × 10^−8)^ and vascular invasion (HR = 1.74, *P* = 0.039).	
Zheng et al. 2018 [16]	To predict postoperative recurrence and survival.	212	107		HR: 2.387 [1.321–4.310]	HR: 3.236 [1.416–7.407]
Kim et al. 2018 [17]	To predict survival with TACE (pretreatment CT).	88		-	The combined model was a better predictor of survival (HR 19.88; *p* < 0.0001).	
Zhou et al. 2017 [18]	To predict the early recurrence (≤1 year) of HCC.	215	No		AUC of 0.82 [0.76–0.87], sensitivity of 0.79, and specificity of 0.70.	NA
					The AUC of the combined model was 0.84 [0.78–0.88], with the sensitivity being 0.82 and specificity 0.71.	
Yang et al. 2022 [19]	To predict MVI status.	198	85		AUC of 0.909, accuracy of 96.47%, sensitivity of 90.91%, specificity of 97.30%, positive predictive value of 83.33%, and negative predictive value of 98.63% in the testing cohort.	
Liu et al. 2021 [20]	To estimate MVI preoperatively.	216	93		AUC: 0.98, Accuracy: 0.95, Sensitivity 0.91, Specificity: 0.97	AUC: 0.82, Accuracy: 0.68, Sensitivity 0.96, Specificity: 0.56
Xu et al. 2022 [21]	To develop a novel nomogram to predict MVI and patients' prognosis based on radiomic features of contrast-enhanced CT.	295	126		AUC of 0.793 (0.714–0.874)	AUC of 0.750 (0.666–0.834)
Liu et al. 2021 [22]	To investigate the predictive value of computed tomography radiomics for MVI in solitary HCC ≤5 cm.	124	61		The radiomics model exhibited a better correction and identification ability in the training and validation groups [area under the curve: 0.72 (95% confidence interval: 0.58–0.86) and 0.74 (95% confidence interval: 0.66–0.83), respectively].	
Zhao et al. 2022 [23]	To investigate the influence of different region of interest (ROI) sizes on CT-based radiomics model for MVI prediction in HCC	In the training set, the sensitivity, specificity, and area under the curve (AUC) of OROI were 0.759, 0.806, and 0.855, respectively. The AUC values of Plus2 (0.979) and Plus3 (0.954) were higher than that of OROI. The AUC values of Plus1 (0.802), Plus4 (0.792), and Plus5 (0.774) were not significantly different from those of OROI. In the validation set, the sensitivity, specificity, and AUC value of OROI were 0.640, 0.630, and 0.664, respectively. The AUC value of Plus3 was 0.903, which was higher than that of OROI. The AUC values of Plus1 (0.679), Plus2 (0.536), Plus4 (0.708), and Plus5 (0.757) were not significantly different from that of OROI (*P* > 0.05).				
Cozzi et al. 2016 [24]	To predict local response and OS treated with VMRT	138	No		Model 1 energy *p* < 0.05, AUC 0.66 [0.56–0.77]	NA
					Model-2 GLNU *p* < 0.05, AUC 0.64 [0.53–0.75]	
					After elastic net regularization, with only compacity significant to Cox model fitting, AUC = 0.80	
Yi-Quan et al. 2021 [25]	To predict MVI preoperatively	110	110		0.980 (CI 0.959–0.993)	0.906 (CI 0.821–0.960)

PT-RO = peritumoral radiomics; T-RO = tumoral radiomics; PT-E = peritumoral enhancement; AUC = area under the curve; OR = odd ratio; PFS=progression free survival; OS = overall survival; DFS = disease free survival; HR = hazard ratio, VMRT = Volumetric-modulated arc therapy.

**Table 3 cancers-15-00743-t003:** Performance of radiomics approaches (using MR features to predict patient outcome).

Author/Year	Study Objectives	Training Set Sample Size	Validation Set Sample Size		Performance (Training Set) [95%CI]	Performance (Validation Set) [95%CI]
Zhang et al. 2019 [26]	To predict early recurrence Gadoxetic acid-enhanced MR (1-year follow-up).	108	47		AUC: 0.844 [0.769–0.919]	
Chen et al. 2022 [27]	To develop and validate radiomics scores and a nomogram of gadolinium ethoxybenzyl-diethylenetriamine pentaacetic acid enhanced MRI for preoperative prediction of MVI in sHCC.	94	100		The AUC of HBP was 0.979, 0.970, and 0.803, respectively, and the AUC of DWI was 0.971, 0.816, and 0.801 (*p* < 0.05), respectively. Good calibration and discrimination of the radiomics and clinical combined nomogram model were exhibited in the testing and two external validation cohorts (C-index of HBP and DWI were 0.971, 0.912, 0.808, and 0.970, 0.843, 0.869, respectively).	
Chen et al. 2021 [28]	To determine the best model for predicting MVI of HCC using conventional gadolinium-ethoxybenzyl-diethylenetriamine pentaacetic acid (gadoxetate disodium)-enhanced MRI features and radiomics signatures with machine learning.	188	81		ADC value, non-smooth tumor margin, and 20-minute T1 relaxation time showed diagnostic accuracy with AUC values of 0.850, 0.847, and 0.846, respectively (*p* < 0.05 for all).	
Kim et al. 2019 [29]	To predict the early and late recurrence of single HCC gadoextic acid-enhanced MR (<2 years vs. >2 years).	128	39		Combined clinicopathologic-radiomic model with 3-mm border extension showed highest c-index: 0.716 [0.627–0.799]; clinicopathologic model: 0.696 [0.557–0.799].	
Hui et al. 2018 [30]	To predict early recurrence (730 days).	50		-	84% accuracy	
Chong et al. 2021 [31]	To predict preoperative MVI and RFS.	230		99	C-indices of 0.700 (0.638–0.763)/C-indices of 0.673 (0.570–0.776) AUCs: 0.920 (0.861–0.979)	

AUC = area under the curve; CI = confidence interval. MVI = microvascular invasion. RFS = recurrence-free survival.

**Table 4 cancers-15-00743-t004:** Performance of radiomics approaches (CT underlying pathology).

Author/Year	Study Objectives	Training Set Sample Size	Validation Set Sample Size		Performance (Training Set) [95%CI]	Performance (Validation Set) [95%CI]
Liao et al. 2019 [32]	To associate with CD8+ T cells	100	42		AUC 0.751 [0.656–0.846]	AUC 0.705 [0.547–0.863]
Ni et al. 2019 [33]	To diagnose MVI.	148	58		The AUCs of the 21 methods ranged from 0.63 to 0.88.	
Mokrane et al. 2019 [34]	To diagnose HCC in cirrhotic patients with indeterminate liver nodules.	142	36		AUC: 0.70 [0.61–0.80]	AUC: 0.66 [0.64–0.84]
Xu et al. 2019 [13]	To associate with MVI.	495		-	AUC: 0.909 in training/validation.	AUC: 0.889 (test setting).
Bakr et al. 2017 [35]	To associate with MVI.	28		-	Slight to moderate agreement (Cohen's kappa range: 0.03 to 0.59)	
Ma et al. 2019 [36]	To associate with MVI.	110	47		C-indices: 0.827	C-indices: 0.820
Peng et al. 2018 [37]	To associate with MVI	184	120		C-index 0.846 [0.787–0.905]	C-index 0.844 [0.77–0.915]

MVI = microvascular invasion; AUC = area under the curve; CI = confidence interval.

**Table 5 cancers-15-00743-t005:** Performance of radiomics approaches (MR underlying pathology).

Author/Year	Study Objectives	Training Set Sample Size	Validation Set Sample Size	Performance (Training Set) [95%CI]	Performance (Validation set) [95%CI]
Gao et al. 2019 [38]	To associate with pathological grading (non-contrast MR).	125	45	AUC: 0.909	AUC: 0.800
Wu et al. 2019 [39]	To differentiate HCC and hepatic hemangioma (non-contrast MR).	295	74	AUC: 0.86	AUC: 0.89
Chen et al. 2019 [40]	To associate with immuno-score in HCC (with Gd-EOB-DTPA MR).	150	57	The combined radiomics-based clinical model AUC: 0.926 [0.884–0.967] The combined radiomics model AUC: 0.904 [0·855–0·953].	Confirmed
Wu et al. 2019 [41]	To associate the grade of HCC with non-contrast-enhanced MR.	125	45	Clinical factor AUC: 0.600 Radiomics signatures AUC: 0.742 the combined clinical and radiomics signature AUC: 0.800	

Gd-EOB-DTPA = gadolinium ethoxybenzyl-diethylenetriaminepentaacetic acid; AUC = area under the curve; CI = confidence interval.

**Table 6 cancers-15-00743-t006:** Performance of radiomics approaches (US underlying pathology).

Author/Year	Study Objectives	Training Set Sample Size	Validation Set Sample Size	Performance (Training Set) [95%CI]	Performance (Validation set) [95%CI]
Hu et al. 2019 [42]	To associate with MVI in HCC (contrast-enhanced ultrasound).	341	141	AUC: 0.731 [0.647, 0.815]	
Yao et al. 2018 [43]	To diagnose HCC and predict PD-1, Ki67, and MVI.	177		AUC: 0.94 [0.88-0.98] for benign and malignant classification, AUC: 0.97 [0.93–0.99] for malignant subtyping, AUC: 0.97 [0.89–0.98] for PD-1 prediction, AUC: 0.94 [0.87–0.97] for Ki-67 prediction, and AUC: 0.98 [0.93–0.99] for MVI prediction.	

MVI = microvascular invasion; AUC = area under the curve; CI = confidence interval.

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
