# Peer review of "Performance of Radiomics in Microvascular Invasion Risk Stratification and Prognostic Assessment in Hepatocellular Carcinoma: A Meta-Analysis"

_cancers, 2023, doi:10.3390/cancers15030743_

Round 1

Reviewer 1 Report

This is an interesting and statistically well conducted meta-analysis.

I have only a major consideration:

You considered articles published to February 2021, so you do not considered works published in the last two years.

Your manuscript will probably be published in 2023, so you have to update the collection of manuscripts for analysis.

Author Response

Reviewer n°1:

Comments and Suggestions for Authors

This is an interesting and statistically well conducted meta-analysis.

I have only a major consideration:

You considered articles published to February 2021, so you do not considered works published in the last two years.

Your manuscript will probably be published in 2023, so you have to update the collection of manuscripts for analysis.

We would like to thank the reviewer for her/his contribution and constructive comments.

We are in complete agreement. We have integrated into the work the articles of the last two years that are part of this study.

In particular, we have integrated the following studies:

Yang et al. 2022 [19]

Liu et al. 2021 [20]

Xu et al. 2022 [21]

Liu et al. 2021 [22]

Zhao et al. 2022 [23]

Yi-Quan et al. 2021 [25]

Chen et al. 2022 [27]

Chong et al. 2021 [31]

Chen et al. 2021 [28]

We have included the resulting changes in the article.

Reviewer 2 Report

Cancers-2128209

Authors submit a meta-analysis of prognostic value of imaging phenomics in HCC. Among the 127 published studies to February 2021 that they extracted using as MESH: Imaging Data Extraction, Radiomics, HCC and usual exclusion criteria, 27 were include for analysis after assessment of bias, modeling (mixed-effect model) and pooling. Providing AUC > 0.79 in training and validation datasets and both they concluded that imaging phenomics is an effective solution to predict prognosis or treatment response in patients with HCC.

In fact, the study seems to be focused on microvascular invasion (MVI) prediction used as kind surrogate marker of patient outcome under several treatments. If right, this point has to be clarify from the title toward M&M up the conclusion.

Comments:

1/ If the assessment of prognostic value of imaging phenomics was limited to its ability to predict MVI clearly state it. Accordingly, title like Performance of radiomics in predicting MVI in HCC patients would be more in line with the content of the paper.

2/ If MVI was the “outcome measure” of the study does it mean that pathological analysis of tumors belonged to selection criteria?

Author Response

Reviewer n°2:

Authors submit a meta-analysis of prognostic value of imaging phenomics in HCC. Among the 127 published studies to February 2021 that they extracted using as MESH: Imaging Data Extraction, Radiomics, HCC and usual exclusion criteria, 27 were include for analysis after assessment of bias, modeling (mixed-effect model) and pooling. Providing AUC > 0.79 in training and validation datasets and both they concluded that imaging phenomics is an effective solution to predict prognosis or treatment response in patients with HCC.

In fact, the study seems to be focused on microvascular invasion (MVI) prediction used as kind surrogate marker of patient outcome under several treatments. If right, this point has to be clarify from the title toward M&M up the conclusion.

We would like to thank the reviewer for her/his contribution and constructive comments.

Comments:

1/ If the assessment of prognostic value of imaging phenomics was limited to its ability to predict MVI clearly state it. Accordingly, title like Performance of radiomics in predicting MVI in HCC patients would be more in line with the content of the paper.

  • We fully agree with your comment. We made the modifications, including also the remarks of reviewer n°3:

Title: Performance of radiomics in microvascular invasion risk stratification and pronostic assessment in hepatocellular carcinoma: a meta-analysis

We have also provided these details in the abstract:

“Imaging phenomics is an effective solution to predict microvascular invasion risk, prognosis and treatment response in patients with HCC.”

Introduction:

“In this regard, the objective of this article was to assess the value of imaging phenomics for microvascular invasion risk stratification and prognostication of HCC.”

 “material and method” section:

“We performed a meta-analysis of studies on prognostic value of imaging phenomics in microvascular invasion risk stratification and prognostication of hepatocellular carcinoma”

And conclusion:

“Imaging phenomics is an effective solution to predict microvascular invasion risk, prognosis and treatment response in patients with HCC.”

2/ If MVI was the “outcome measure” of the study does it mean that pathological analysis of tumors belonged to selection criteria?

  • That's right. We brought in this missing information in the inclusion criteria:

“We identified all studies that assessed the prognostic value of imaging phenomics in hepatocellular carcinoma. Studies were eligible for inclusion if they (1) were retrospective or prospective; (2) assessed imaging phenomics; (3) reported pathological analysis of tumors including MVI information; (4) reported the association with cancer prognosis or response to treatment.”

Reviewer 3 Report

Prognostic value of imaging phenomics in hepatocellular carcinoma: a meta-analysis

Sylvain Bodard, Yan Liu, Sylvain Guinebert, Kherabi Yousra, Tarik Asselah

The main goal of this meta-analysis is to assess the value of imaging phenomics for risk stratification and prognostication of Hepatocellular carcinoma (HCC). The authors have performed a meta-analysis of manuscripts published to Febryary 2021 on MEDLINE and have showed that imaging phenomics is an effective solution to predict prognosis or diagnosis in patients with HCC.

The manuscript in part is well structured and written but there are some points that need corrections.

The following are some general considerations.

First of all, the title is misleading: it focuses on the prognostic value while the declared goal is to assess the value of imaging phenomics for risk stratification and prognostication of HCC. In my opinion it should be modified by including both terms.

Secondly, sections 2.1 and 2.2 of the materials and methods seem to me to be unclearly written in that, for example, exclusion criteria are stated in both while they should be listed only in 2.2. The search strategy should have clear indications of the period and databases used (giving rationale for choosing a single database).

Figure 1 appears to be incomplete, missing some information necessary for a proper flow diagram. I suggest to authors follow what is suggested in the PRISMA guidelines. See Page MJ, McKenzie JE, Bossuyt PM, Boutron I, Hoffmann TC, Mulrow CD, et al. The PRISMA 2020 statement: an updated guideline for reporting systematic reviews. BMJ 2021;372:n71. doi: 10.1136/bmj.n71

Finally, some information should be added in the results regarding the prediction interval and distribution derived from the analysis performed. These two parameters should then be discussed in the discussion along with  any limitations of the evidence included in the review and of the review processes used.

Minor corrections are:

carrying out the acronym HCC in line 13;

the addition of spaces before the brackets of references;

adding a bullet point in “......were analysed Our study......” in line 25;

using the lowercase character in line 120 and 131;

some typos in figure 2 captions.

Author Response

Dear Referees,

We would like to thank the reviewers and the editorial board for their constructive feedbacks. We modified the text and the figure significantly in order to address their criticism. In the process, we believe the article was improved and made more precise. The editor and reviewers will find below our response for each of the points they raised.

Reviewer n°3:

The main goal of this meta-analysis is to assess the value of imaging phenomics for risk stratification and prognostication of Hepatocellular carcinoma (HCC). The authors have performed a meta-analysis of manuscripts published to Febryary 2021 on MEDLINE and have showed that imaging phenomics is an effective solution to predict prognosis or diagnosis in patients with HCC.

The manuscript in part is well structured and written but there are some points that need corrections.

We would like to thank the reviewer for her/his contribution and constructive comments.

The following are some general considerations.

First of all, the title is misleading: it focuses on the prognostic value while the declared goal is to assess the value of imaging phenomics for risk stratification and prognostication of HCC. In my opinion it should be modified by including both terms.

  • We fully agree with your comment. We made the modifications including also the remarks of the reviewer n°2:

Title: Performance of radiomics in microvascular invasion risk stratification and pronostic assessment in hepatocellular carcinoma: a meta-analysis

We have also provided these details in the abstract:

“Imaging phenomics is an effective solution to predict microvascular invasion risk, prognosis and treatment response in patients with HCC.”

Introduction:

“In this regard, the objective of this article was to assess the value of imaging phenomics for microvascular invasion risk stratification and prognostication of HCC.”

 “material and method” section:

“We performed a meta-analysis of studies on prognostic value of imaging phenomics in microvascular invasion risk stratification and prognostication of hepatocellular carcinoma”

And conclusion:

“Imaging phenomics is an effective solution to predict microvascular invasion risk, prognosis and treatment response in patients with HCC.”

Secondly, sections 2.1 and 2.2 of the materials and methods seem to me to be unclearly written in that, for example, exclusion criteria are stated in both while they should be listed only in 2.2. The search strategy should have clear indications of the period and databases used (giving rationale for choosing a single database).

  • We have removed the exclusion criteria from part 2.1 where they were effectively out of place.
  • We have specified our choice to refer to only one database

“We have chosen to focus the study on this database because of the many high-quality articles referenced and easily accessible.”

Figure 1 appears to be incomplete, missing some information necessary for a proper flow diagram. I suggest to authors follow what is suggested in the PRISMA guidelines. See Page MJ, McKenzie JE, Bossuyt PM, Boutron I, Hoffmann TC, Mulrow CD, et al. The PRISMA 2020 statement: an updated guideline for reporting systematic reviews. BMJ 2021;372:n71. doi: 10.1136/bmj.n71

  • We have modified Figure 1 by adding the missing information as suggested in the PRISMA guidelines:

(cf. doc)

Finally, some information should be added in the results regarding the prediction interval and distribution derived from the analysis performed. These two parameters should then be discussed in the discussion along with  any limitations of the evidence included in the review and of the review processes used.

  • We have made these clarifications and limitations in the discussion

“The prediction interval and distribution (0.80 (95% CI 0.76-0.84)) derived from the analysis are acceptable in our analysis. However, our study is limited to the MEDLINE database, which may be a limitation, although it offers many quality articles.”

Minor corrections are:

carrying out the acronym HCC in line 13;

  • We have made the correction.

the addition of spaces before the brackets of references;

  • We have made the correction.

adding a bullet point in “......were analysed Our study......” in line 25;

  • We have made the correction.

using the lowercase character in line 120 and 131;

  • We have made the correction.

some typos in figure 2 captions.

  • We have made the correction.

Round 2

Reviewer 1 Report

Thank you for updating the references.

I have no more commentaries.

Reviewer 2 Report

Authors made substantiate changes according to reviewers’ comments. However, since they clearly stated that radiomics work was focused on its ability to predict MVI the conclusion “Imaging phenomics is an effective solution to predict microvascular invasion risk, prognosis and treatment response in patients with HCC” remains too broad:

MVI is robust prognostic marker in HCC patients undergoing surgical treatments (ie transplantation and resection). The prognostic value of this biomarker for patients receiving systemic or locoregional (like TACE) has never be investigated. In the field of ablative treatments (like RFA), extrapolation of the prognostic value of MVI could be conceptually acceptable even if definitive proof will never exist (statement of presence or absence of MVI request pathological examination of the whole tumor surgically removed). Thus, predicting MVI, the radiomics tool developed by authors could assess the outcome and the prognosis of HCC patients treated with transplantation and resection and likely ablation but not treatments in general.